# Biological and Histological Assessment of the Hepatoportoenterostomy Role in Biliary Atresia as a Stand-Alone Procedure or as a Bridge toward Liver Transplantation

**DOI:** 10.3390/medicina57010016

**Published:** 2020-12-28

**Authors:** Raluca-Cristina Apostu, Vlad Fagarasan, Catalin C. Ciuce, Radu Drasovean, Dan Gheban, Radu Razvan Scurtu, Alina Grama, Ana Cristina Stefanescu, Constantin Ciuce, Tudor Lucian Pop

**Affiliations:** 1Department of Surgery, “Iuliu Hatieganu” University of Medicine and Pharmacy Cluj-Napoca, 8 Victor Babes Street, 400000 Cluj-Napoca; First Surgical Clinic, Emergency County Hospital, 3-5 Clinicilor Street, 400006 Cluj-Napoca, Romania or ralucaapostu@ymail.com (R.-C.A.); vlad.fagarasan@yahoo.com (V.F.); catalinciuce@gmail.com (C.C.C.); radu.drasovean@umfcluj.ro (R.D.); cciuce@umfcluj.ro (C.C.); 2Department of Pathology, “Iuliu Hatieganu” University of Medicine and Pharmacy Cluj-Napoca, 8 Victor Babes Street, 400000 Cluj-Napoca; 4 th Pediatric Clinic, Emergency Clinical Hospital for Children, 68 Motilor Street, 400000 Cluj-Napoca, Romania; dan.gheban@umfcluj.ro; 3Department of Pediatrics, “Iuliu Hatieganu” University of Medicine and Pharmacy Cluj-Napoca, 8 Victor Babes Street, 400000 Cluj-Napoca; 2nd Pediatric Clinic, Emergency Clinical Hospital for Children, 400177 Cluj-Napoca, Romania; gramaalina16@yahoo.com (A.G.); anacristinastefanescu@gmail.com (A.C.S.); tudor.pop@umfcluj.ro (T.L.P.)

**Keywords:** biliary atresia, hepatoportoenterostomy, cirrhosis, transplant, survival

## Abstract

*Background and objectives*: In patients with biliary atresia (BA), hepatoportoenterostomy (HPE) is still a valuable therapeutic tool for prolonged survival or a safer transition to liver transplantation. The main focus today is towards efficient screening programs, a faster diagnostic, and prompt treatment. However, the limited information on BA pathophysiology makes valuable any experience in disease management. This study aimed to analyze the evolution and survival of patients with BA referred for HPE (Kasai operation) in our department. *Materials and Methods*: A retrospective analysis was performed on fourteen patients with BA, diagnosed in the pediatric department and further referred for HPE in our surgical department between 2010 and 2016. After HPE, the need for transplantation was assessed according to patients cytomegalovirus (CMV) status, and histological and biochemical analysis. Follow-up results at 1–4 years and long term survival were assessed. *Results*: Mean age at surgery was 70 days. Surgery in patients younger than 60 days was correlated with survival. Jaundice’s clearance rate at three months was 36%. Total and direct bilirubin values had a significant variation between patients with liver transplants and native liver (*p* = 0.02). CMV was positive in eight patients, half with transplant need and half with native liver survival. Smooth muscle actin (SMA) positivity was proof of advanced fibrosis. The overall survival rate was 79%, with 75% for native liver patients and an 83% survival rate for those with liver transplantation. Transplantation was performed in six patients (43%), with a mean of 10 months between HPE and transplantation. Transplanted patients had better survival. Complications were diagnosed in 63% of patients. The mean follow-up period was six years. *Conclusions*: HPE, even performed in advanced cirrhosis, allows a significant survival, and ensures an essential time gain for patients requiring liver transplantation. A younger age at surgery is correlated with a better outcome, despite early CMV infection.

## 1. Introduction

Biliary atresia (BA) is one of the pediatric diseases still under palliative treatment in most cases, with an overall survival that has improved to 80–90% [1]. Surgery is the first step in this management but its efficiency depends on a quick diagnosis, so it might be considered an emergency procedure. Although many alternative variants to hepatoportoenterostomy (HPE, Kasai operation) have been proposed, they were not able to replicate the same results [2]. This was a reason to change the focus toward a faster diagnostic, given that younger patients age have a less advanced stage of the disease, with consecutively, better results. The ideal time for surgery is currently considered between the first 30–45 days of life, although when performed at 45–60 days it seemed to be associated with better results [3,4]. An important tool for the early detection of BA would be a screening program in which the efficiency of stool color cards is highlighted [5] together with case centralization in high volume centers [6].

Nowadays, long-term survival means that patients can reach up to 20-years of age and more [7]. However, even with the best experience, there are still undetermined processes in the pathophysiology of the disease leading to a high risk of complications which finally require liver transplantation in 30% to 50% of patients [8]. However, an early timing may be associated with higher operative risks for the small-sized recipients and side-effects of prolonged immunosuppression, while a prolonged waiting time may increase the risk of complications on advanced cirrhosis. Therefore, a personalized decision should take into consideration the donor availability, the technical expertise and also the parental decision, and therefore also including the Kasai operation in case management [9].

We aimed to analyze the evolution and survival of patients with BA referred for Kasai surgery in our department, in correlation with the patients’ age, Metavir score and CMV infection at surgery, their biochemical parameters and jaundice clearance. Postoperative complications and the need for liver transplantation have been also evaluated in correlation with the above mentioned parameters.

## 2. Materials and Methods

We performed a retrospective analysis of patients with BA admitted between 2010 and 2016 in the 1st Surgical Clinic and the Emergency Clinical Hospital for Children. The study was conducted in accordance with the rules of the Helsinki Declaration. Fourteen patients in whom BA was confirmed were referred for HPE in our department. In all cases, the perinatal/acquired form was identified, with jaundice persisting for more than 2 weeks from birth, acholic stools, dark urines, hepatomegaly, weight deficiency, and high bilirubin levels. The cytomegalovirus (CMV) infection was verified by IgM and IgG assays in all cases. The BA diagnosis was established by correlating their clinical picture with ultrasonography. In five patients, with still uncertain ultrasonographic findings, we performed a magnetic resonance cholangiopancreatography (MRCP) as well. Hepatic fibrosis was assessed using transient elastography (Fibroscan) and graded according to the Metavir score.

Exploratory surgery was performed in all cases by the same surgical team, led by Professor Dr. Constantin Ciuce. A bisubcostal incision allowed access in the peritoneal cavity. The liver was mobilized and the presence of the gallbladder was checked. The fibrotic remnant was dissected and detached from the liver, advancing towards the hilum, where it was ligated and sectioned, while the hepatic arteries and the confluence of the portal vein branches were identified. The hilar plate was exposed and all fibrous tissue excised. Bile ductules were identified in the sectioned area. A jejunal loop was prepared at 15 cm below the Treitz ligament and advanced through the mesocolon towards the hepatic hilum. A termino-lateral HPE was performed, using resorbable sutures in a Roux-en-Y fashion. The entero-enteral anastomosis was performed at 40 cm distally from the HPE. Additionally, a hepatic biopsy was performed. The cavity was washed and a subhepatic drain was left, followed by laparorraphy.

The excised tissue and the biopsy were analyzed, using hematoxylin-eosin and trichrome Masson staining. Alpha-smooth muscle actin (SMA) and Cytokeratin 7 (CK7) positivity were assessed by immunohistochemistry. 

Postoperatively all patients received antibiotics, ursodeoxycholic acid, corticosteroids, and fat-soluble vitamins.

Biochemical analysis was performed before and after surgery. Follow-up was performed at 1, 3, 6, and 12 months and afterward annually or when complications occurred. The recorded values were compared with those from the literature. Depending on the evolution, patients were evaluated regarding the need for liver transplantation. A comparative analysis between the transplanted and native liver patients was also performed. The results were analyzed by t-test, nonparametric correlation test, ANOVA for variance analysis, Levene test for variations between groups, and Kaplan–Meier survival analysis, using R and IBM SPSS programs.

## 3. Results

All patients presented with persistent jaundice for more than 6 weeks of life, the average age for surgery reference being 10 weeks (6–28 weeks). The average weight at presentation was 5 kg (2.8–7 kg). An equal distribution by gender was observed. At ultrasonography, in all patients, the gallbladder was absent or difficult to identify, with no patent main bile duct. In five patients an additional MRCP was used to confirm the diagnosis and showed the absence of extrahepatic bile ducts and gallbladder, associated with hepatomegaly and ascites. 

Before surgery, the Metavir score was F4 in eight patients (57%), F3 in four patients (29%) and F2 in two patients (14%). The score was confirmed by the intraoperative aspect of the liver, micronodular, cirrhotic in all cases and the postoperative histopathological analysis of the hepatic parenchyma. Four years after surgery 75% of F4 patients had to be transplanted while 25% remained stationary; also, in all the remaining patients the Metavir score advanced by a point: F3 to F4 and F2 to F3, respectively. 

The mean age at surgery was 70 days (42 to 196 days). Four patients underwent surgery at more than 60 days of life (29%) and one patient was operated at 196 days. Ten patients were operated at less than 60 days (71%), 2 of these patients at less than 45 days. We identified a significant relationship between age of fewer than 60 days at surgery and survival (*p* = 0.02) with no correlation found between survival rate and jaundice clearance verified at 1, 3, 6, and 12 months (*p* = 0.34; *p* = 0.92; *p* = 0.55; *p* = 0.85). 

The length of hospital stay (LHS) in the surgical department was between 6 and 18 days, with a mean LHS of 11 days.

The mean value of the preoperative total bilirubin was 13 mg/dL ± 5.32 (7–23.01 mg/dL). The mean value of the direct bilirubin was 10 mg/dL ± 4.36 (5.89–19 mg/dL) while the gamma-glutamyl-transferase (GGT) mean value was 798 U/L ± 564.50(84–1653 U/L). There was no statistical correlation between these values and the patients’ gender (*p* = 0.66; *p* = 0.63; *p* = 0.79). GGT values were normal before surgery in two patients (14%).

Postoperative mean values of total and direct bilirubin at 1, 3, 6, and 12 months respectively are summarized in Table 1. Five patients (36%) had a total bilirubin value <2 mg/dL at three months. Postoperative jaundice clearance rate at 1, 3, 6, and 12 months are represented in Table 2. No significant difference was noted between total bilirubin values at 1, 3, 6, and 12 months after surgery (*p* = 0.48). A statistically significant difference was identified for the jaundice clearance rate at 6 months when compared to values registered at 1 month after surgery (*p* = 0.005).

Total and direct bilirubin values had a significant variation between patients with transplant and patients with native liver after the Kasai operation (*p* = 0.02). However, after surgery at 1, 3, 6, and 12 months no significant differences were noted in values between these two groups (*p* = 0.11; *p* = 0.24; *p* = 0.36; *p* = 0.28; *p* = 0.34; *p* = 0.54; *p* = 0.37; *p* = 0.59). 

The analysis of the portal fibrotic tissue excised during surgery identified porta hepatis with rare bile ducts of medium caliber, with fibrous bridges. The fibrous septa presented in the central area irregular, collapsed bile ducts, with pan CK positive epithelium and in the periphery, ductular proliferation, similar to the embryonic ductal plate (Figure 1).

The main histological changes identified in the hepatic parenchyma were hepatic lobules with irregular nodular organization, thick porto-portal fibrous bridges, and neutrophilic and lymphocytic inflammatory infiltrate. Moreover, regenerative nodules were identified, with no centro-lobular veins, surrounded by bridging fibrosis. Marked ductular proliferation was noticed peripherally, with a plexiform distribution in the central area of the bridges. Marked cholestasis, with ductal bile plugs both in the periphery and in the interhepatocyte ducts was a characteristic aspect.

SMA was intensely positive on septa around the biliary neoducts. Some of the remaining normal liver lobules had veins, with sinusoids in which Ito stellate cells produced SMA, the cells being intensely positive. SMA positivity was also identified in Kupffer and endothelial cells with deposits in the Disse spaces; also, intensity for SMA in the capillaries was higher when they were closer to the confluence with the centro-lobular vein. In the portal structures, SMA was positive in the muscle layer of the arteries and the normal bile ducts (Figure 2).

All patients with native liver were screened for complications. Cirrhosis, cholangitis, portal hypertension with esophageal varices grade I and hypersplenism were diagnosed in 63% of them. Cholangitis episodes were recurrent, with a higher frequency in the first year after surgery and all cases were managed conservatively.

Liver transplantation was necessary for eight patients (57%), including the two patients who had surgery at less than 45 days. Six patients were transplanted (43%) in specialised centers. All of them had split liver transplants performed, from their relatives and the immunosuppressive therapy was managed according to those transplantation centers protocols. The average period between HPE and transplant was 10 months (3 months–1.8 years). Transplantation was performed at a median age of one year (6 months–2 years). The survival rate for the transplanted patients at one and two years was 100%, while at three and four years it was 83%.

The overall survival rate was 79%. The survival rate for patients with their native liver at one year was 82%, while at two, three, and four years it was 75%. For patients who underwent surgery before the age of 60 days and survived with the native liver, the survival rate was 60% (six patients) while for those with surgery between 60 and 90 days of age, the survival rate was 33% (one transplanted patient). A significant difference was noticed between the transplanted and the native liver group (*p* = 0.004) (Figure 3). 

When BA was diagnosed, eight patients had IgM for CMV. Of these, four patients required a transplant and four survived with native liver. A statistically significant difference was recorded in survival between CMV infected patients and the negative ones, in favor of the positive group (*p* = 0.04).

The mortality rate was 21% (three patients). Two patients died with pulmonary complications and one patient due to cirrhosis complications. 

Patients had a median follow-up of six years (4.5–10 years). The oldest transplanted patient had a follow-up of 10 years, while the oldest patient with native liver was followed up for 8 years.

## 4. Discussion

In BA a quick diagnosis is crucial for the patients’ evolution. The persistence of high direct and total bilirubin values over the first two weeks of life imposes a specialized evaluation. The early diagnosis depends on several factors, such as the existence of a screening program, regular postnatal medical surveillance, patient education. For clinical screening, the use of stool cards has been an efficient method to reduce the age at diagnosis [5,8,10].

Other ultrasonographic, laboratory, and histopathological parameters have been defined as part of a scoring system, able to discriminate between BA and other causes of neonatal cholestasis [11]. In addition to ultrasonography, the role of other diagnostic imaging methods has been evaluated, the use of MRCP, ERCP, or PTCC having a limited role [6]. In our center, the ultrasonographic reports were adapted to the standardized protocols only for the last patients included in this study. For this reason, the only constant parameters identified were phantom gallbladder and the absence of a patent main bile duct. In five cases the ultrasonographic findings were uncertain and, while waiting for the exploratory laparotomy, we decided to perform a MRCP, as a noninvasive method to imagistically confirm the diagnosis.

In our study, the mean age of 70 days at surgery, can be explained by the absence of a screening program for neonatal cholestasis and a lack of an active follow-up of neonates in their first two months of life. In our group of patients, except for a patient operated at 196 days of age, the mean age at surgery was 58 days (42–84 days), which is concordant with the European guidelines [6]. Even though 60 days is considered a cut-off value for a successful surgery, the best results are thought to be obtained in patients aged between 45 and 60 days [3]. Some studies suggested surgery may be performed even before the age of 30 days, achieving jaundice clearance in all patients [7]. However, other studies are showing a worse prognostic in those cases [12]. For CMV-associated form, a successful surgery window time was suggested to be in the age interval of 41 to 60 days old [4].

In our group of patients, 71% of them underwent HPE before the age of 60 days, while 14% were younger than 45 days. Patients of a younger age at HPE required liver transplantation, one year later, while in those operated after 60 days of life, 75% survived with their native liver. All these patients survived at least till puberty. It would seem that the optimal interval for achieving the best results after HPE would be 45–60 days.

Of the remaining four patients that underwent surgery after 60 days of age, only one survived, with liver transplantation, performed three months after HPE, while for the others the death occurred through pulmonary or cirrhosis complications. Data from the literature showed that at an age of 60–90 days, the biliary drainage is efficient in 40–50% of cases, while over 90 days, this rate drops to 25% and at less than 20% when surgery is performed over 100 days [13]. When surgery is performed at 121–150 days of life biliary drainage occurs only in 7.7% of patients [7]. However, there are some studies with 5 and 10-year survival rates of 45% and 40% even though HPE was performed in patients aged over 100 days [14]. 

The need for an earlier diagnostic has triggered the search for possible biomarkers that could be determined after birth and guide the diagnostic process. Such markers are direct bilirubin level, GGT values, and direct/total bilirubin ratio.

Even though the difference between the mean values of total and direct bilirubin recorded in the present study, were not statistically significant between genders, we identified some possible predictive cut-off values worth considering as indicators for transplantation. We identified a preoperative cut-off value of 7.2 mg/dL which was predictive for transplant indication, while six months after surgery values less than 2 mg/dL were correlated with survival with the native liver. Reported cut-off values in the literature were assessed at one week after surgery and a total bilirubin value under 4.85 mg/dL was correlated with native liver survival and jaundice clearance [15]. 

The elevation of direct bilirubin levels within the first days after birth was also suggested to be a screening instrument for biliary atresia [5,16]. Guidelines suggest that a direct bilirubin level over 1 mg/dL is a signal for further evaluation and referral to a hepatologist [6]. We only had available direct bilirubin levels at more than two weeks after birth and preoperatively and they did not differ significantly between the transplanted patients and those who survived with their native liver. 

GGT is also thought to be an important biomarker for the disease evolution. There are reports on a subgroup of patients with normal GGT values at diagnosis and having apparently a worse outcome compared to the others. These patients register a quicker need for liver transplantation after HPE [17]. In our group, two patients presented normal GGT values and had an evolution marked by complications. In one patient surgery was performed at the age of 196 days and he died two months later of cirrhosis complications. In the other patient, surgery was performed at 56 days of age and was complicated with anastomotic fistula and transplantation five months later. Since a patient presented late in the course of the disease while another one presented on time, both with normal GGT values and both having an evolution marked by complications, it is probable that GGT value would be suitable for consideration as a marker for advanced or aggressive disease. 

A better outcome after surgery was predicted by a jaundice clearance at three months [18]. We recorded 36% of cases with clearance in whom one patient required transplantation, while the others survived with their native liver. Despite the absence of clearance at three months after HPE, 43% of patients survived with their native liver. 

Even though, there is a proven correlation between the biliary clearance rate and patient survival, in our group, the patients survived for a long term despite a lack of clearance at three months. This result might be due to the small number of patients in our study. This could also be proof that despite a continuous evolution of the disease, this process is slowed down by the HPE. A further argument that might support this is based on the Metavir score analysis. Most of the F4 patients were transplanted two years after HPE, but after four years 25% of them showed no progressive disease while those with lower grades slowly advanced during the postoperative follow-up.

SMA is a useful immunohistochemical marker for the development of liver fibrosis and a predictive tool for early liver fibrogenesis in infants with biliary atresia. The hepatic expression is a histopathological marker for activated stellate cells responsible for liver fibrogenesis [19]. In our study, SMA was intensely positive in the bridges around neoducts, expressed by the Ito stellate cells, but also Kupffer and endothelial cells, as proof of advanced fibrosis.

CMV infection in BA is associated with a poor outcome and a specific subtype of BA is defined as being CMV-associated. The diagnosis relies on liver biopsies with positive staining for IgM against CMV. These patients were reported to have the highest mortality [1]. Positive patients may present with a later onset, the infection being the cause for a delayed referral of patients [20]. This could explain a more severe liver fibrosis in this subgroup [21]. The reported incidence ranges between 34% and 69% [22]. In our study, 57% of patients were CMV positive. Half of the positive patients required liver transplantation, so their evolution cannot be considered as being favourable after the HPE, explaining better survival due to transplantation. The other half survived with their native liver and benefited from the favorable effects of HPE, which slowed down the BA evolution. Moreover, the serum positivity was identified associated with isolated disease, which led us to consider that these patients could have been infected after birth and thus having better survival when compared to the rest of the group. We also found that the isolated form of infection was not associated with the worst prognosis. Some similar reports were unable to identify immunohistochemical evidence of the virus in the liver or biliary tissue in their IgM positive group, suggesting an inactive state of infection, while other studies showed no difference in the postoperative outcome independently of the CMV infection [15,21,22]. In long-term outcome, no significant differences were registered [20], as our study confirmed, all CMV positive patients being registered as survivals. 

HPE is considered a palliative treatment, however, it is associated with improved survival rates. The most promising results were reported by a Japanese group, with 5-, 10-, and 20-years survival rates of 63%, 54%, and 44%, while Wong et al reported a 51% survival rate at 20 years [23]. In Europe, a survival rate of up to 30% at 20 years was reported [24]. At 1 to 3 years the reported survival rates with the native liver are between 20.3% and 75.8% [10]. In our study, survival in the first year was higher, 82%, while from the second year it became 75% and remained constant. The reported patient survival after transplantation was 90% at 6 months and 88% at 3 years [25].

Early complications are rare. In our study, only one patient had an anastomotic leak that was solved surgically on the eighth postoperative day. One of the most frequent complications is cholangitis. Modifications of the procedure suggested to avoid this complication have not been able to overcome the initial technique [2]. Early cholangitis is associated with adverse outcomes and recurrence leads to liver failure and severe complications. Patients with cholangitis have lower overall survival rates [26,27]. In our study, deceased patients had early severe complications, while 83% of the patients with their native liver and long-term survival had cholangitis.

A mortality rate of 21% was registered in our study. One patient died two months after surgery from cirrhotic complications. In this case, the surgery was performed at an age of 196 days. Another patient with surgery at 70 days, had biliary clearance at three months, but died at 11 months from pulmonary complications, with normal bilirubin values. The third patient had surgery at 63 days, but no jaundice clearance at three months, was transplanted at 1.8 years after HPE and died because of post-transplantation complications. Before the age of two years, 57% of patients from our study required transplantation, a value similar to those from the literature [9]. The period between surgery and transplantation was less than two years in our study.

The management of biliary atresia patients is multidisciplinary and the best results are published from high volume centers. Our personal experience is similar to a low volume center and was presented in order to compare results with more experienced ones and to improve efforts for early diagnosis and proper management.

## 5. Conclusions

The importance of HPE cannot be underestimated since it ensures the survival of about half of the patients with BA. Early diagnosis ensures prompt treatment and has an important role in slowing down the disease. Even in the absence of a jaundice clearance at three months after HPE, patients might have long-term survival with their native livers, provided that complications are efficiently managed. Although the HPE results are better in high volume centers, a dedicated team with an experienced surgeon might reach comparable rates of long term survival. Patients younger at the time of surgery seemed to be significantly correlated with a better outcome, while no significant differences in long-term outcome were recorded with regard to early CMV infection. Until liver transplantation will be highly accessible or new treatments or prophylactic methods will become available, HPE remains a useful tool in BA management.

## Figures and Tables

**Figure 1 medicina-57-00016-f001:**
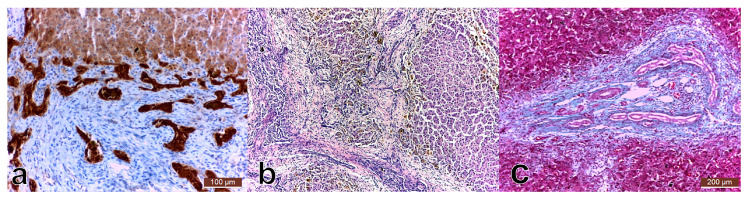
Hepatic biopsy: (**a**). Bile ductular proliferation (CK7 (cytokeratin), 20×, 100 µm); (**b**). Liver nodules with no centrolobular vein, with intracellular and canalicular cholestasis (HE (hematoxylin eosin), 100×); (**c**). Ductal plate malformation (TM (trichrome masson), 10×, 200 µm).

**Figure 2 medicina-57-00016-f002:**
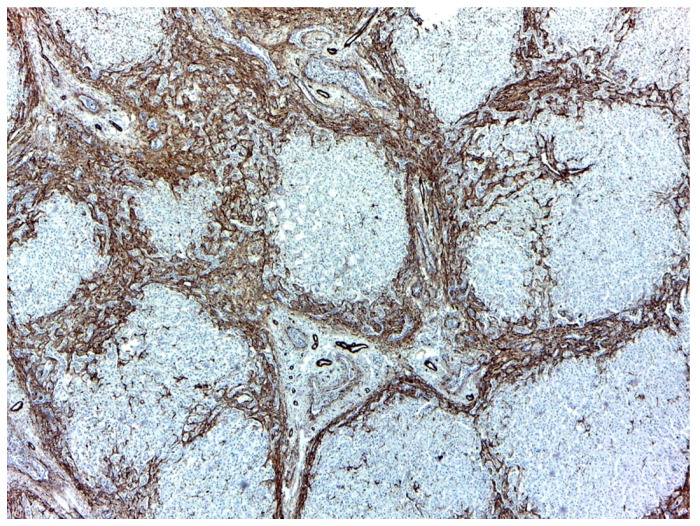
Smooth muscle actin (SMA) highly positive on the bridges around the neoformed bile ducts. Overall appearance 40×.

**Figure 3 medicina-57-00016-f003:**
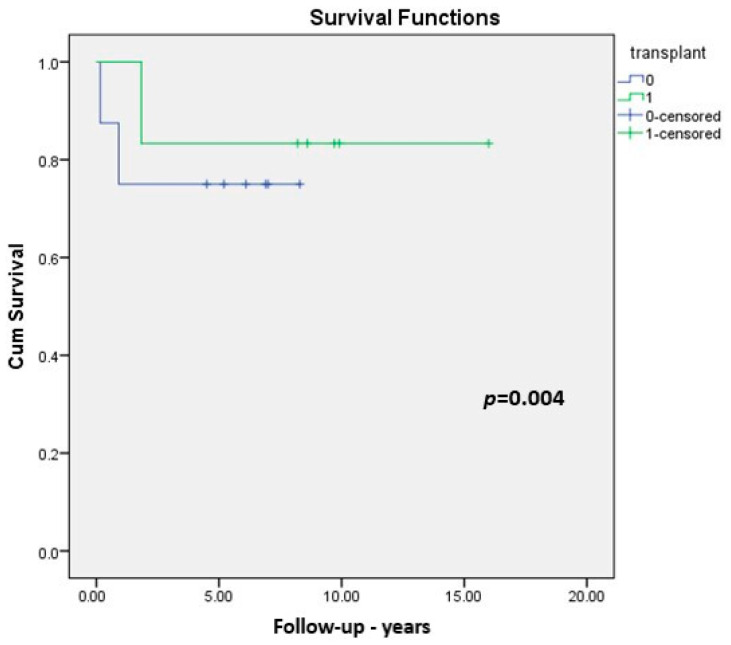
Survival for transplanted and native liver patients.

**Table 1 medicina-57-00016-t001:** Mean values of total and direct bilirubin.

Marker	Time Interval after Surgery			
	1 Month	3 Months	6 Months	12 Months
TB (mg/dL) Mean ± SD	10.1 ± 8.63	10.4 ± 11.65	9.33 ±12.19	7.52 ± 8.03
DB (mg/dL) Mean ± SD	7.77 ± 6.74	7.49 ± 8.06	6.93 ± 8.81	5.85 ± 6.31

TB—total bilirubin, DB—direct bilirubin, SD—standard deviation.

**Table 2 medicina-57-00016-t002:** Clearance rate after surgery (TB > 2 mg/dL).

Marker	Time Interval after Surgery			
*n* = 14 Patients	1 Month	3 Months	6 Months	12 Months
Clearance 0 (TB > 2 mg/dL)	78.57%	64.28%	50%	71.42%
Clearance 1 (TB < 2 mg/dL)	21.43%	35.72%	50%	28.58%

TB—total bilirubin.

## Data Availability

The data presented in this study are available on request from the corresponding author. The data are not publicly available due to ethical and privacy issues involving the subjects.

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
