# Peer review of "Biological and Histological Assessment of the Hepatoportoenterostomy Role in Biliary Atresia as a Stand-Alone Procedure or as a Bridge toward Liver Transplantation"

_medicina, 2020, doi:10.3390/medicina57010016_

Round 1
Reviewer 1 Report
This retrospective study analyzed the evolution and survival of 14 patients with biliary atresia (BA) after hepatoportoenterostomy (Kasai operation) observed in a single department.
Unfortunately, no original information was provided. Actually, there are many studies, available in the literature and also included in the references of Apostu's study, that have already studied the medium and long-term evolution of BA children undergoing Kasai on larger series. Even the predictors of survival with native liver have been well investigated so it is difficult to understand what Apostu's study adds if not the disclosure of a limited local experience. Furthermore, the baseline ultrasound data appears very poor. There is no indication to the parameters of El Guindi et al (Design and validation of a diagnostic score for biliary atresia. J Hepatol 2014; 61(1):116-23). Authors should also clarify why magnetic resonance cholangiopancreatography (MRCP) was used in 5 cases considering that the guidelines do not recommend its routine use (JPGN 2017; 64(1):154-168). According to guidelines, limited specificity of MRCP, ERCP, PTCC provides a limited role in the general guidance to caregivers toward diagnosing BA in the present era.
Reviewer 2 Report
I recommend improving the information concerning the transplant: type of graft (cadaveric pediatric full-size? Split? LRLT?) What type of immunosuppressions?
Reviewer 3 Report
I read the manuscript written by Apostu and colleagues. The authors reported their single-center experience on hepatoportoenterostomy in biliary atresia patients.
In 6 years, they reported 14 cases for the present study. Several reported experiences with larger cohorts are reported in the literature.
Different suggestions:
The aims of the study are not correctly reported looking at the results section.
The importance and role of CMV infection at diagnosis need to be investigated and developed.
The difference in survival between CMV + and CMV - needs to be correlated with the liver transplantation. Eight patients underwent OLT, from the CMV+ group?
Photos are too many, please marge to 1 or max two photos.
Round 2
Reviewer 3 Report
I read the revised manuscript. The authors have reply to all my suggestions and improved the overall manuscript. As reported, considering the low number of cases, all experiences can add elements to the clinical practive.